# A Density-Dependent Modified Doraivelu Model for the Cold Compaction of Poly (Ether Ketone Ketone) Powders

**DOI:** 10.3390/polym14061270

**Published:** 2022-03-21

**Authors:** Fan Xu, Huixiong Wang, Xuelian Wu, Zihao Ye, Hong Liu

**Affiliations:** 1School of Mechanical Engineering, Jiangsu University, Zhenjiang 212000, China; xlwu@ujs.edu.cn (X.W.); qq379556433@icloud.com (Z.Y.); 1000004949@ujs.edu.cn (H.L.); 2School of Mechanical Engineering and Automation, University of Science and Technology Liaoning, Anshan 114051, China

**Keywords:** PEKK powder, cold compaction, modified Doraivelu model, UMAT

## Abstract

The cold compaction of poly (ether ketone ketone) (PEKK) powder was studied by experiments and simulations based on the modified Doraivelu model. Although this model can successfully predict the compaction behavior of metal powders, discussion of the prediction of polymer powders is lacking. Based on the mechanical theory of metal plasticity, the modified Doraivelu model was established by introducing the material parameters *m* and *n*. The modified model can predict the compaction density of PEKK powder during cold compaction. A sub-increment method for this constitutive model was then established and implemented into a finite-element model by using the user-defined material subroutine UMAT in ABAQUS/Standard. Consequently, the material parameters of the modified Doraivelu model were identified by an inverse method using the experimental data and simulation results. It was found that when *m* = 0, *n* = 4, and the initial relative density was 0.4485, the simulation results were the closest to the experimental ones.

## 1. Introduction

Powder cold-compaction technology is extensively used in the molding process of metals, ceramics, pharmaceutical materials, and composites owing to the advantages of near net molding. Its molding theory and technology are also relatively mature [1,2,3,4,5,6]. In terms of polymer materials, cold-compaction technology has overcome the difficulties of injection molding and extrusion molding for special plastics, such as ultra-high-molecular-weight polyethylene, polytetrafluoroethylene, or polyvinylidene chloride owing to high viscosity [7,8,9]. Compared with hot-compaction technology, cold compaction is more energy-saving, has no heating and cooling stage, and has a shorter molding period. Owing to uncertain factors, such as external friction and compaction conditions, studying the densification behavior of plastic powder under cold compaction is significant to prevent and control product failure.

Previous studies have primarily focused on the densification behavior of powders in cold compaction through compression equation, the continuous-mechanics method, and the mesoscopic model [10,11,12]. The most representative compression equations are the Balshen, Kawakita–Lüdde, Huang [13], and Heckel [14] equations. Although these equations can reveal the mechanical behavior of powder materials, they lack the ability to predict density distribution. Meso-simulation is used to model each particle, and its calculation accuracy is related to the number of particles. Conversely, the actual number of compacted particles is huge, resulting in a substantial amount of calculation [15]. Moreover, the actual powder particles are often assumed as spheres in the microscopic simulation, which leads to large errors in the analysis [16]. To improve the accuracy, numerous experimental calibration-model parameters are needed, which increases the computational cost. Hu et al. [7] scanned and analyzed polymer–crystal composite particles by X-ray microcomputed tomography and scanning electron microscopy (SEM). They obtained the real shape of polymer-crystal composite particles and accurately predicted the densification behavior of polymer–crystal composite materials during cold compaction by mesoscopic simulation. The prediction accuracy was higher than that of the sphere model, but it required a series of complicated parameter-calibration and model-verification experiments, and the calculation was huge. Thus, the industrial application of this simulation work is difficult to realize.

The CAM-clay model [17], Cap model [18], and Drucker-Prager Cap (DPC) model [19] in continuum mechanics have been designed to discuss the mechanical behavior of rock and soil particles. However, because rock and soil particles are similar to powder particles, they are applied in the numerical analysis of powder compaction. Among them, the DPC model [19] is attracting the attention of many scholars because it can describe the densification and hardening behavior of powders. However, owing to the large number of parameters and complex determination methods, this model cannot be used extensively in simulation research. Shima-Oyane [20], Gurson [21], and Doraivelu [22] et al. proposed an ellipsoidal yield criterion based on the mechanical theory of metal plasticity. Doraivelu et al. [22] hypothesized that the physical interpretation of the von Mises yield criterion is based on the hypothesis that metal powder enters a plastic state when the total elastic specific energy is constant. Compared with the yield conditions of Shima [20], Gurson [21], and others, this hypothesis has a better theoretical basis. In other words, the parameters have clear physical properties and are easy to obtain through experiments. The simulation results are also consistent with the experimental ones [22], which is widely recognized [23,24,25]. Additionally, the Doraielu model has been revised to expand the scope of the application of the model.

In the present study, a poly (ether ketone ketone) (PEKK) sample was processed by powder cold compaction, and its densification behavior was studied. Based on the von Mises yield criterion combined with the Doraivelu powder constitutive model, the material constants *m* and *n* were proposed to enhance the expression ability of the original model in terms of density and to expand the model’s application scope. The numerical calculation of the model was studied, and the radial-regression algorithm was used to solve the model and complete the derivation process of the algorithm. Furthermore, the user subroutine UMAT provided by ABAQUS solver was used to write source code in Fortran language and complete the numerical simulation. Combined with the cold-compaction process of PEKK powder, the effect of the influence law of the initial relative density on the prediction accuracy of the model was revealed to further verify the model’s reliability.

## 2. Establishing the Constitutive Model of PEKK Powders

### 2.1. PEKK Powder Performance

PEKK, as an important material in national defense and the military industry, has excellent characteristics, such as high bearing capacity, high wear resistance, high heat resistance, and high oil resistance [26,27]. In this study, homemade PEKK powder was studied by cold compaction. The PEKK powder was observed with an Olympus optical digital microscope (DSX500, Shanghai Liyang Industrial Co., Ltd, Shanghai, China), and the micromorphology of the PEKK particles was examined. As shown in Figure 1, the average particle size was about 35 μm. In addition, a differential scanning calorimeter (DSC) (STA449-C, NETZSCH, Free State of Bavaria, Germany) test was carried out. As shown in Figure 2, $T_{g}=169.3$ °C and $T_{m}=360$ °C, which is extremely close to the products of OXPEKK series [28]. Because of the limited conditions of the research group, we adopted other physical properties of OXPEKK products, made minor corrections, and found that the density of the PEKK was 1.35 g/cm^3^, the yield strength was 138 MPa and the elastic modulus was 4500 MPa.

### 2.2. Modified Doraivelu Constitutive Model

Theoretically, the influences of porosity and hydrostatic pressure on porous materials need to be considered, so the yield criterion can generally be expressed as Equation (1) [13].
(1)$$f\left(\sigma_{ij}\right)=AJ_{2}^{\prime }+BJ_{1}{}^{2}=\eta \sigma_{s}{}^{2}$$
where $J_{2}^{\prime }$ and $J_{1}$ are the second invariant of the deviatoric stress tensor and the first invariant of the stress tensor, respectively; $\sigma_{s}$ is the yield stress of the material in the dense state; and η is a function of relative density and represents the contribution of geometric hardening. *A* and *B* must meet the following conditions.
(2)$$\left\{\begin{array}{l} \underset{\rho \to 1}{\lim}A=3\\ \underset{\rho \to 1}{\lim}B=0\\ \underset{\rho \to 1}{\lim}\eta =1 \end{array}\right.$$

In other words, when the compact was in a fully dense state, it degenerated into the von Mises yield criterion.

*A* and *B* are the coefficients of the yield function and can be expressed as functions of Poisson’s ratio. Doraivelu et al. [22] derived the relationship between *A*, *B* and Poisson’s ratio based on the assumption of elastic specific energy, as shown in Equation (3).
(3)$$A=2\left(1+\nu \right),B=\frac{1-2\nu }{3}$$

The elastic modulus and Poisson’s ratio should be considered as functions of relative density. Doraivelu et al. [22] adopted the empirical formula of Poisson’s ratio and relative density proposed by Zhdanovich [29],
(4)$$\nu =\frac{1}{2}\rho^{2}$$

Based on the results of dimensional analysis, Equation (5) of the flow-stress factor was proposed as follows:(5)$$\eta =\frac{\rho^{2}-\rho_{c}{}^{2}}{1-\rho_{c}{}^{2}}$$
where $\rho_{c}$ is the critical relative density of porous material state without flow stress, which was obtained through experiments and close to the initial relative density. Thus, the Doraivelu constitutive model can be obtained as follows:(6)$$(2+\rho^{2})J_{2}^{\prime }+\frac{1-\rho^{2}}{3}J_{1}{}^{2}=\frac{\rho^{2}-\rho_{c}{}^{2}}{1-\rho_{c}{}^{2}}\sigma_{s}{}^{2}$$

Wang et al. [30] discussed the compaction behavior of iron powder. They proved that the linear Equation (7) of Poisson’s ratio and relative density had higher prediction accuracy.
(7)ν=0.93ρ−0.43

Substituting Equation (7) into Equation (3) yields Equation (8)
(8)$$A=1.86\rho +1.14,B=0.62\left(1-\rho \right)$$

Meanwhile, Wang et al. scaled the coefficient *B* to obtain Equation (9)
(9)$$B=0.32\left(1-\rho \right)$$

However, the yield coefficient *B* at this time did not conform to Equation (3). Kim et al. [25] believed that the selection of parameters *A* and *B* primarily depends on experimental data. Although Tszeng and Wu [24] emphasized in their study that porous materials must strictly comply with uniaxial stress conditions, evidence that this condition also applies to loose powders is lacking. Song et al. [31] also supported Kim et al.‘s point of view, and used the models *A* and *B* proposed by Wang et al. [30] to revise the flow-stress model (Equation (10)). They regarded the elastic modulus as a linear function of relative density (Equation (11)) and found that the revised model has good predictive ability.
(10)$$\eta =\frac{\rho^{7}-\rho_{c}{}^{7}}{1-\rho_{c}{}^{7}}$$
(11)$$E=E_{s}\rho$$

According to the linear equation of Poisson’s ratio and relative density of iron powder compaction [30], and based on the yield coefficient *B* and the flow-stress model [31], the constitutive model of Doraivelu was modified as follows:(12)$$(1.86\rho +1.14)J_{2}^{\prime }+m(1-\rho )J_{1}{}^{2}-\frac{\rho^{n}-\rho_{c}{}^{n}}{1-\rho_{c}{}^{n}}\sigma_{s}{}^{2}=0$$
where *m* and *n* are material constants, i.e., $m\;\epsilon \;\left[0,0.62\right]$ and *n* is a real number greater than 1.

### 2.3. Numerical Implementation of Stress-Integration Algorithm

The yield condition cannot be directly used in elastoplastic finite-element analysis, but it can be used to deduce the elastoplastic constitutive relation of materials. Elastoplastic deformation is a nonlinear mechanical problem generally solved by the Newton–Raphson iterative method to establish the constitutive relation between stress increment and strain increment. The stress and relative density need to be updated iteratively.

During the compaction process, the density of the powder changed constantly, and the density was closely related to the accumulated bulk plastic strain. With increased compaction time, the accumulated bulk plastic strain and density constantly increased. Density and cumulative volumetric plastic strain must satisfy the law of conservation of mass:(13)$$\rho +\Delta \rho =\frac{\rho_{0}l_{1}l_{2}l_{3}}{\left(l_{1}+\Delta l_{1}\right)\left(l_{2}+\Delta l_{2}\right)\left(l_{3}+\Delta l_{3}\right)}=\frac{\rho_{0}l_{1}l_{2}l_{3}}{l_{1}l_{2}l_{3}\exp(\varepsilon_{ij}^{p}\delta_{ij})}=\rho_{0}\exp(-\varepsilon_{ij}^{p}\delta_{ij})$$
(14)$$d\rho =-\rho d\varepsilon_{ij}^{p}\delta_{ij}$$

Assuming that the stress was updated to the next incremental step and the strain increment was a purely elastic process, it was determined by Hooke’s law as follows:(15)σt+Δttrial=σt+Δσtriale=σt+DeΔε
$D_{e}\;$is the elastic stiffness matrix. The elasticity of the strain increment was determined by Equation (12). If $\left(1.86\rho +1.14\right)J_{2trial}^{\prime }+m\left(1-\rho \right)J_{1trial}{}^{2}-\frac{\rho^{n}-\rho_{c}^{n}}{1-\rho_{c}^{n}}\sigma_{s}{}^{2}<0$, it was an elastic process, and σt+Δt=σt+Δttrial,ρt+Δt=ρt. Otherwise, it was an elastoplastic process, and the elastic trial stress crossed the yield surface. The stress in the yield surface was updated by the elastic matrix, and the stress outside the yield surface was updated by the elastoplastic matrix. Therefore, it was necessary to calculate the ratio of the elastic strain increment and plastic strain increment to the total strain increment. Let the ratio of elastic strain increment to total strain increment be $\alpha \left(0<\alpha <1\right)$. Subsequently, the value of α was obtained by dichotomy, and the solution flow chart is shown in Figure 3.

After obtaining the value of *α*, the stress and relative density were updated to the elastoplastic boundary
(16)σ_t+Δt=σt+Δttrial+αDeΔε
(17)ρ_t+Δt=ρt

Considering the high nonlinear degree of the elastoplastic process, the Euler integration method has only second-order accuracy [32], which may result in large error accumulation for the elastoplastic process with a high degree of nonlinearity. Accordingly, we used the substepping method for the constitutive integration of the elastoplastic process, that is, the total elastoplastic strain increment was divided into several sub-increments. The key aspect of the substepping method was to determine the sub-increment step. If the sub-increment step was too small, it led to a large cumulative error. If the sub-increment step was too large, it sacrificed the operational speed. Thus, we applied the stress-explicit integration method. This method can automatically control the error [32,33] and size of the sub-increment step according to the nonlinear degree of the elastoplastic process, taking into account the calculation accuracy and efficiency. The specific analysis is shown in Figure 4.

Where $D^{ep}$ is the elastoplastic stiffness matrix, $T_{0}\in \left(0,1\right]$, $\delta_{ij}=\left\{\begin{array}{c} 0\;i\neq j\\ 1\;i=j \end{array}\right.\;i,j=1,2,3$, $TOL=10^{-6}$, $f\left(\sigma ,\rho \right)$ is expressed by Equation (12), and *k* is the plastic-correction coefficient. According to the consistency condition, plastic modification should be performed after updating the stress in each substep so that the renewed stress still falls on the yield surface. 

In this study, corrections were made along the plastic-flow direction [34], and the plastic-correction coefficient is shown in Equation (18).
(18)$$k=\frac{f(\sigma_{i},\rho )}{\left(\partial f/\partial \sigma )(\partial g/\partial \sigma \right)}$$

Considering that this work was based on the assumption of associated flow law, the plastic potential was equal to the yield function; thus, Equation (18) can be rewritten as
(19)$$k=\frac{f(\sigma_{i},\rho )}{{\left(\partial f/\partial \sigma \right)}^{2}}$$

The derivation of the elastoplastic matrix $D^{ep}$ and the rewriting of Equation (12) were as follows
(20)$$f=AJ_{2}^{\prime }+BJ_{1}{}^{2}+C=0$$
where A=1.86ρ+1.14, B=m(1−ρ), and $C=-\frac{\rho^{n}-\rho_{c}^{n}}{1-\rho_{c}^{n}}\sigma_{s}^{2}$.

From the above equation, it is possible to obtain
(21)$$\frac{dA}{d\rho }J_{2}^{\prime }d\rho +A\frac{\partial J_{2}^{\prime }}{\partial \sigma_{ij}}d\sigma_{ij}+\frac{dB}{d\rho }J_{1}{}^{2}d\rho +2BJ_{1}\frac{\partial J_{1}}{\partial \sigma_{ij}}d\sigma_{ij}+\frac{\partial C}{\partial \rho }d\rho =0$$

Let $H_{A}=\frac{dA}{d\rho }$, $H_{B}=\frac{dB}{d\rho },H_{C}=\frac{dC}{d\rho },$ it is possible to obtain
(22)$$H_{A}J_{2}^{\prime }d\rho +A\frac{\partial J_{2}^{\prime }}{\partial \sigma_{ij}}d\sigma_{ij}+H_{B}J_{1}{}^{2}d\rho +2BJ_{1}\frac{\partial J_{1}}{\partial \sigma_{ij}}d\sigma_{ij}+H_{C}d\rho =0$$

From $d\rho =-\rho d\varepsilon_{ij}^{p}\delta_{ij}$, it is possible to obtain
(23)$$-\rho (H_{A}J_{2}^{\prime }+H_{B}J_{1}{}^{2}+H_{C})d\varepsilon_{ij}^{p}\delta_{ij}+A\frac{\partial J_{2}^{\prime }}{\partial \sigma_{ij}}d\sigma_{ij}+2BJ_{1}\frac{\partial J_{1}}{\partial \sigma_{ij}}d\sigma_{ij}=0$$

According to the hypothesis of the associated flow rule $d\varepsilon_{ij}^{p}=d\lambda \partial f/(\partial \sigma_{ij})$, the elastic stress–strain relationship can be used to obtain
(24)$$d\sigma_{ij}=D_{ijkl}^{e}d\varepsilon_{kl}^{e}=D_{ijkl}^{e}\left(d\varepsilon_{kl}-d\varepsilon_{kl}^{p}\right)=D_{ijkl}^{e}d\varepsilon_{kl}-D_{ijkl}^{e}d\lambda \frac{\partial f}{\partial \sigma_{kl}}$$
where $D_{ijkl}^{e}$ is the elastic stiffness matrix.

Substituting Equation (22) into Equation (21) yields
(25)$$-\rho (H_{A}J_{2}^{\prime }+H_{B}J_{1}{}^{2}+H_{C})d\varepsilon_{ij}^{p}\delta_{ij}+(A\frac{\partial J_{2}^{\prime }}{\partial \sigma_{ij}}+2BJ_{1}\frac{\partial J_{1}}{\partial \sigma_{ij}})\left(D_{ijkl}^{e}d\varepsilon_{kl}-D_{ijkl}^{e}d\lambda \frac{\partial f}{\partial \sigma_{kl}}\right)=0$$

The above equation is arranged to obtain
(26)$$d\lambda =\frac{(A\frac{\partial J_{2}^{\prime }}{\partial \sigma_{ij}}+2BJ_{1}\frac{\partial J_{1}}{\partial \sigma_{ij}})D_{ijkl}^{e}}{\rho (H_{A}J_{2}^{\prime }+H_{B}J_{1}{}^{2}+H_{C})\frac{\partial f}{\partial \sigma_{ij}}\delta_{ij}+(A\frac{\partial J_{2}^{\prime }}{\partial \sigma_{ij}}+2BJ_{1}\frac{\partial J_{1}}{\partial \sigma_{ij}})D_{ijkl}^{e}\frac{\partial f}{\partial \sigma_{kl}}}d\varepsilon_{kl}$$
where $\frac{\partial f}{\partial \sigma_{ij}}=A\frac{\partial J_{2}^{\prime }}{\partial \sigma_{ij}}+2BJ_{1}\frac{\partial J_{1}}{\partial \sigma_{ij}}$.

Substituting into Equation (24) yields
(27)$$d\lambda =\frac{\frac{\partial f}{\partial \sigma_{ij}}D_{ijkl}^{e}}{\rho (H_{A}J_{2}^{\prime }+H_{B}J_{1}{}^{2}+H_{C})\frac{\partial f}{\partial \sigma_{ij}}\delta_{ij}+\frac{\partial f}{\partial \sigma_{ij}}D_{ijkl}^{e}\frac{\partial f}{\partial \sigma_{kl}}}d\varepsilon_{kl}$$

Combining the form of dλ with Equation (22), it is possible to obtain
(28)$$d\sigma_{ij}=(D_{ijkl}^{e}-\frac{D_{ijmn}^{e}\frac{\partial f}{\partial \sigma_{mn}}D_{klrs}^{e}\frac{\partial f}{\partial \sigma_{rs}}}{\rho (H_{A}J_{2}^{\prime }+H_{B}J_{1}{}^{2}+H_{C})\frac{\partial f}{\partial \sigma_{ij}}\delta_{ij}+\frac{\partial f}{\partial \sigma_{ij}}D_{ijkl}^{e}\frac{\partial f}{\partial \sigma_{kl}}})d\varepsilon_{kl}=D_{ijkl}^{ep}d\varepsilon_{kl}$$

## 3. Experiment and Simulation

### 3.1. Experimental Method

The cold-compaction experiment with PEKK powder was performed on a UTM4000 (SUNS, Shenzhen, China) electronic universal testing machine (Figure 5a). The maximum load of this testing machine was 10 kN. The compacting tools are shown in Figure 5b,c. The compacting die and punch have a circular cross-section, and the molding-cavity diameter was 2.5 mm. The force–displacement curve of the top punch was obtained by monitoring the compaction process with the sensor. After adding the PEKK powder into the mold, the punch was prepressed at a compaction speed of 0.1 mm/min. When the punch force reached 100 N, the prepressing process ended, and precompacted compacts were obtained with relative densities of 0.4352, 0.4485 and 0.4377 and pre-compaction heights of 1.921, 2.093 and 2.196 mm, respectively. Next, the force and displacement were set to zero and the compaction speed was kept constant for compaction. The loading was stopped when the force on the top punch reached 2000, 2000 and 2500 N and the force–displacement curves of the three groups of experiments were obtained. At the same time, a group of compacts with low initial relative density and a group of compacts with high initial relative density were compacted, the initial relative densities were 0.2558 and 0.6767, respectively. The loading was stopped when the loading forces reached 500 and 1500 N, respectively. The green compact was removed and measured, and we obtained green-compact heights of 0.888, 0.966, 0.977, 1.171 and 0.840 mm, as well as masses of 5.44, 6.22, 6.37, 5.56 and 5.38 mg. The relative densities of the green compacts were calculated by Equation (29) and the relative densities of the five green compacts were 0.9415, 0.9719, 0.9837, 0.7165 and 0.9667, respectively (Figure 5d). The obtained force–displacement curve is shown in Figure 6. The results of five groups of tests are shown in Table 1. In addition, we observed the surface of the compact with a microscope.
(29)$$\rho =\frac{m}{\pi R^{2}H\rho_{s}}$$
where *ρ* is the relative density of the green compact, *m* is the mass of the green compact, *R* is the radius of the green compact (1.25 mm), H is the height of the green compact, and $\rho_{s}$ is the theoretical density of the PEKK powder (1.35 g/cm^3^).

### 3.2. Cold-Compaction Simulation

The cold-compaction process of the PEKK powder was simulated using ABAQUS2020 software. The modified Doraivelu model proposed served as the constitutive model, whereas the UMAT user subroutine was adopted to realize the model. The stress-updating algorithm was deduced in detail in the second section of this paper. Five groups of samples were simulated. The diameter of cylindrical samples was 2.5 mm, while the initial heights were 1.921, 2.093, 2.196, 3.28 and 1.207 mm, respectively. Assuming that the die was a rigid body, the Coulomb friction model was adopted, the friction coefficient between he PEKK powder and die wall was set to 0.1, and the contact algorithm adopted the penalty-function method. The element type was an eight-node linear hexahedron element C3D8I, and the mesh size was 0.2 mm.

## 4. Results and Discussion

### 4.1. Parameter Identification

(1) Determination of *n*

Compared with the experimental data of the second group, *m* was set 0.62 and we simulated the cold compaction of the powder with an initial height of 2.093 mm when *n* = 2, 4 and 7. The force–displacement curve of the upper punch was obtained and compared with the experiment, as shown in Figure 7 In the late stage of the powder compaction, the force–displacement curves obtained by these three flow-stress models were far from those of the experiment and the error was large. However, in the early stage of compaction, when *n* = 2, the parameter was overestimated in the low-density stage of the powder, which led to difficult powder flow for a long time and hindered the densification process. When *n* = 7, the “softening effect” in the early stage of powder compaction was exaggerated. However, when *n* = 4, the force–displacement curve of the flow-stress model was close to the experimentally measured value, so we determined that the flow-stress model *n* was 4 and that the further value was as follows:(30)$$\eta =\frac{\rho^{4}-\rho_{c}{}^{4}}{1-\rho_{c}{}^{4}}$$

(2) Determination of *m*

Although the flow-stress model can accurately predict the early stage of powder compaction, the constitutive model was poor at predicting the late stage of powder compaction. Thus, the value of *m* was determined next, so that the modified model could accurately predict the entire compaction process. The force–displacement curve of the punch obtained by the simulation with *m* = 0.62, 0.15, 0.04 and 0 was compared with the experimental value, as shown in Figure 8. With decreased *m*, the force-displacement curve approached that of the experiment and, finally, *m* was determined to be 0. Therefore, a modified Doraivelu model applied to the cold-compaction process of PEKK powder is proposed in this paper, as follows:(31)$$(1.86\rho +1.14)J_{2}^{\prime }=\frac{\rho^{4}-\rho_{c}{}^{4}}{1-\rho_{c}{}^{4}}\sigma_{s}{}^{2}$$

### 4.2. Verification of Modified Doraivelu Model

According to the green-compact quality, the average relative density of the powder in different stages of compaction was calculated, and the load was converted to pressure to obtain the compaction equation of the powder. Compared with the simulation results, the relative error between the two was calculated. As shown in Figure 9, when the pressure was 32 MPa, the relative error reached the maximum value of −6.87%, but in the middle and late stages of compaction, when the relative density reached more than 0.7, the maximum error was only 2.9%, which was relatively high.

### 4.3. Influence of Initial Density on the Prediction Accuracy of the Model

The initial relative densities of the five groups of experiments were 0.4352, 0.4485, 0.4377, 0.2558 and 0.6767, respectively. The force–displacement curves of the five groups of experiments and simulations are shown in Figure 10. When the initial relative density was 0.4485, the prediction accuracy was higher, which could better predict the compaction stages. However, when the initial relative density was 0.4352 and 0.4377, the model could better simulate the early stage of compaction. However, the prediction effect of initial relative density of 0.2208 and 0.6767 was poor in the middle and late stages of compaction, although it could still accurately simulate the change in load and displacement at the end of compaction. As shown in Figure 10, the error was larger in each period of compaction. The relative density was calculated using the accumulated plastic strain. Thus, for the sample with the initial relative density of 0.2208, the pores were larger and the rearrangement of the particles and the filling of the pores primarily occurred in most of the compaction period. The plastic deformation was also smaller, which led to larger errors. However, for the sample with an initial relative density of 0.6767, the pores were few. At this point, the green compact had undergone significant plastic deformation in the preloading stage, so it produced many errors when simulating the initial state of compaction.

Figure 11 shows the nephogram of the relative density distribution of the five compacts after compaction. Figure 10 shows that the experimental data of the second group were the closest to the simulation comparison. Figure 11b shows that the density was higher in the top-right corner and lower in the bottom-right corner, which was due to the friction between the mold and the powder. The density distribution in other areas was also relatively uniform. The other four sets of simulation results showed that the density-distribution law was similar to that shown in Figure 11b. The average relative densities of the five green compacts obtained by simulation were 0.9163, 0.9463, 0.9494, 0.7093 and 0.9483, respectively. Compared with the average relative densities of the five green compacts experimentally obtained, i.e., 0.9415, 0.9719, 0.9837, 0.7165 and 0.9667, it was found that they were close to those of the simulation ones. Therefore, although the prediction accuracy of the entire model process was poor at low and high initial relative density, it could still accurately predict the average relative density at the end of compaction.

Figure 12 shows the microstructure of the upper surfaces of five compacts, among which no obvious pores can be seen in Figure 12a–c,e, mainly because the density of these four samples was higher, while obvious pores can be seen in Figure 12d, mainly because the density of this sample was lower. No boundary between particles can be seen in Figure 12, which indicates that the powder particles underwent obvious plastic deformation.

## 5. Conclusions

A modified Doraivelu model for predicting the cold compaction of PEKK powder was proposed.

(1)The modified Doraivelu model based on PEKK powder was developed by introducing the material parameters *m* and *n*. A substepping method for the constitutive model was developed using UMAT in ABAQUS/Standard and applied to the finite-element model.(2)Through experiments and numerical calculations, it was determined that when *m* = 0, *n* = 4 and the initial relative density was 0.4485, the error was the smallest and the prediction accuracy was the highest. Therefore, the modified model can accurately describe the change law of density during cold compaction.

## Figures and Tables

**Figure 1 polymers-14-01270-f001:**
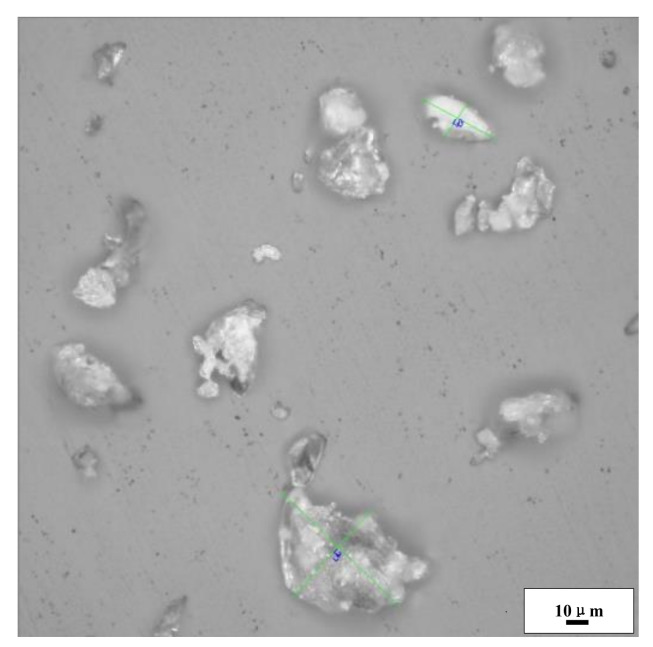
The microgram of PEKK powers.

**Figure 2 polymers-14-01270-f002:**
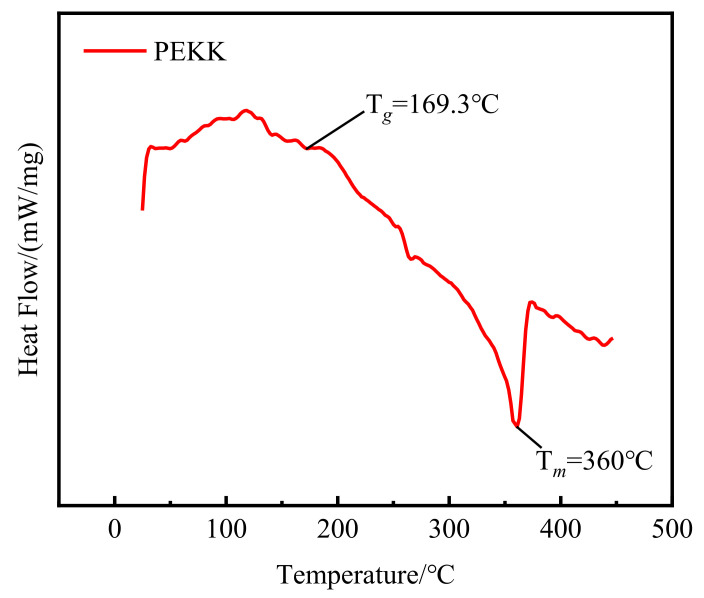
DSC curves of PEKK in the heating.

**Figure 3 polymers-14-01270-f003:**
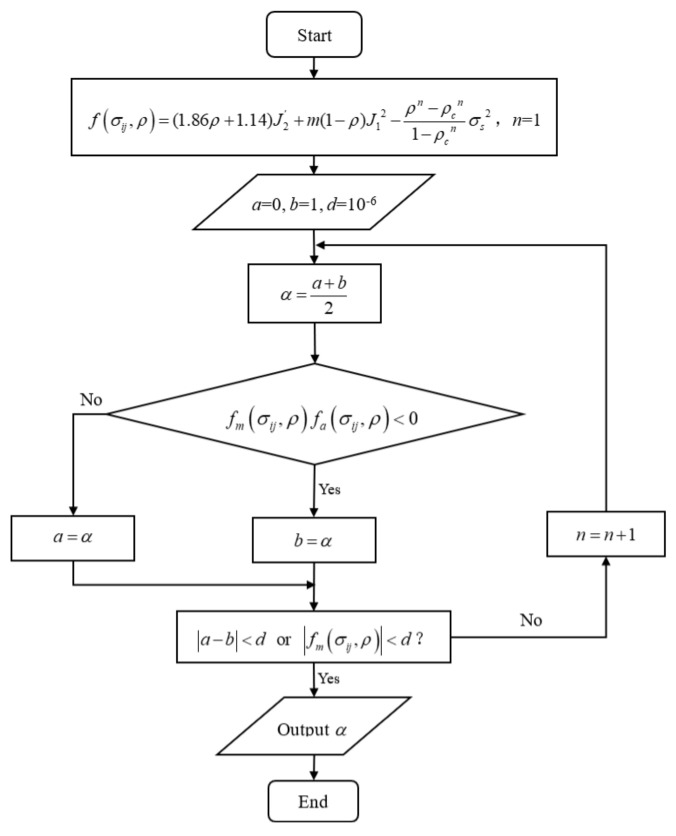
Flow chart of finding *α* by dichotomy.

**Figure 4 polymers-14-01270-f004:**
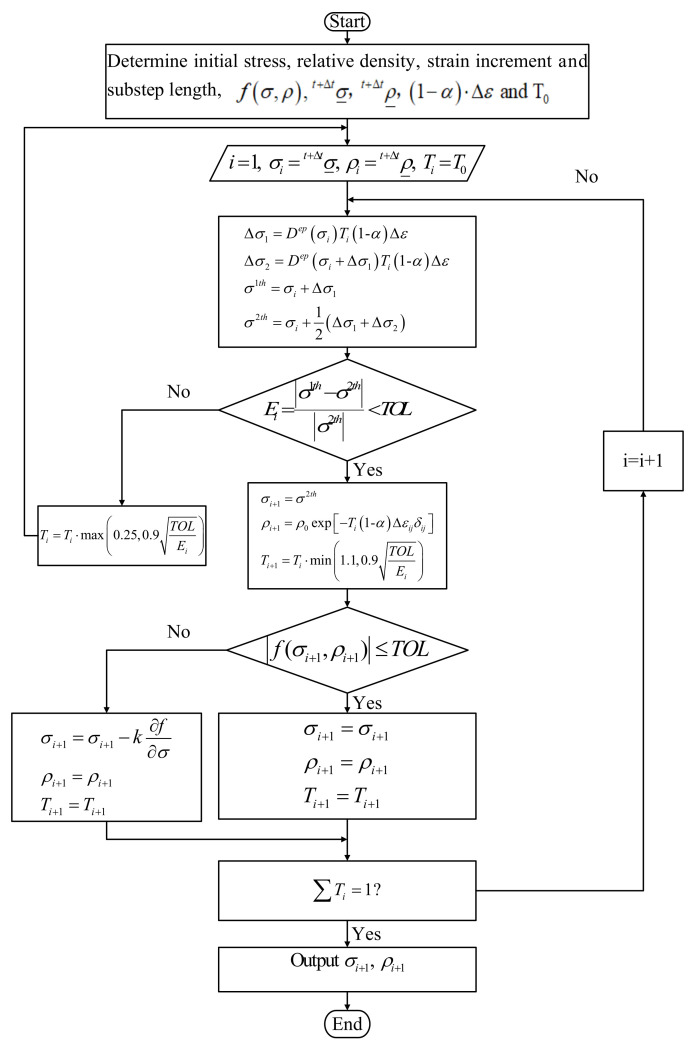
Stress-explicit integration method for automatic control error.

**Figure 5 polymers-14-01270-f005:**
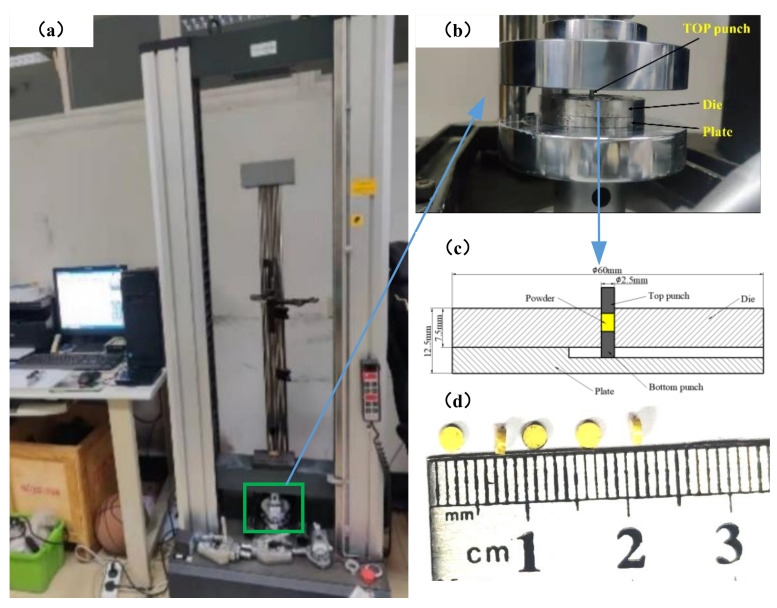
UTM4000 electron universal testing machine. Explanation in the text.

**Figure 6 polymers-14-01270-f006:**
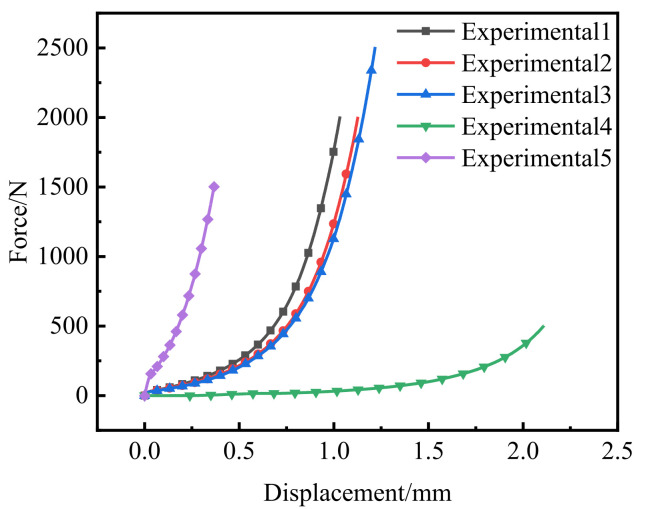
Force–displacement curve.

**Figure 7 polymers-14-01270-f007:**
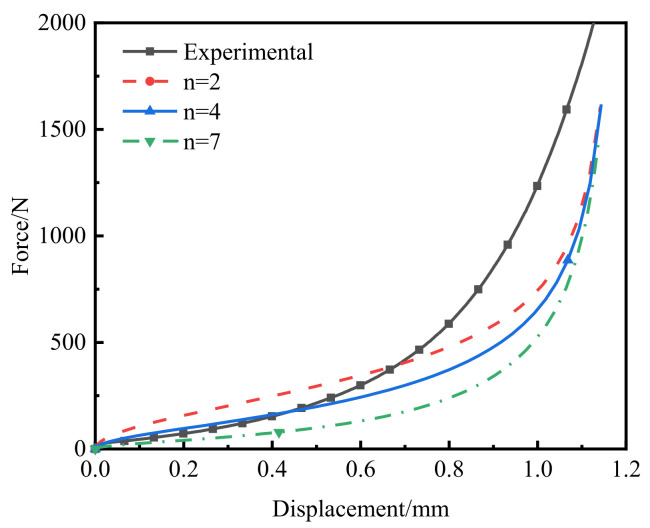
Calibration procedure of *n* value of flow stress model.

**Figure 8 polymers-14-01270-f008:**
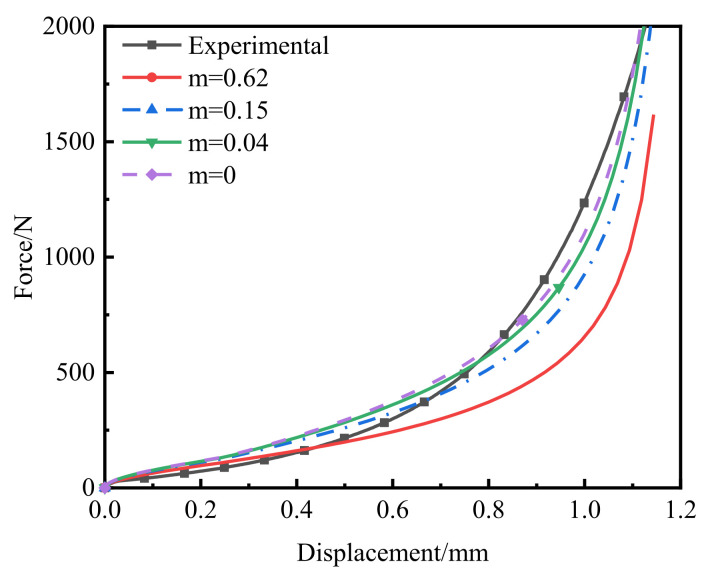
Calibration procedure for *m*.

**Figure 9 polymers-14-01270-f009:**
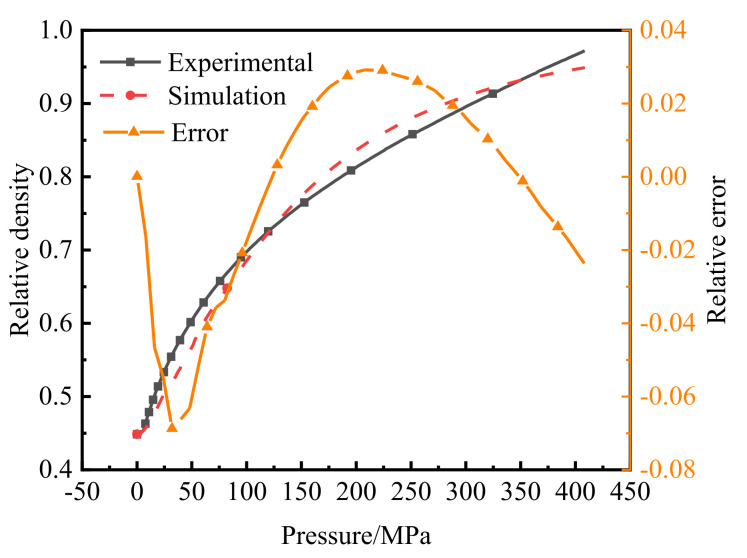
Comparisons of the compaction equation from numerical simulations and experimental results.

**Figure 10 polymers-14-01270-f010:**
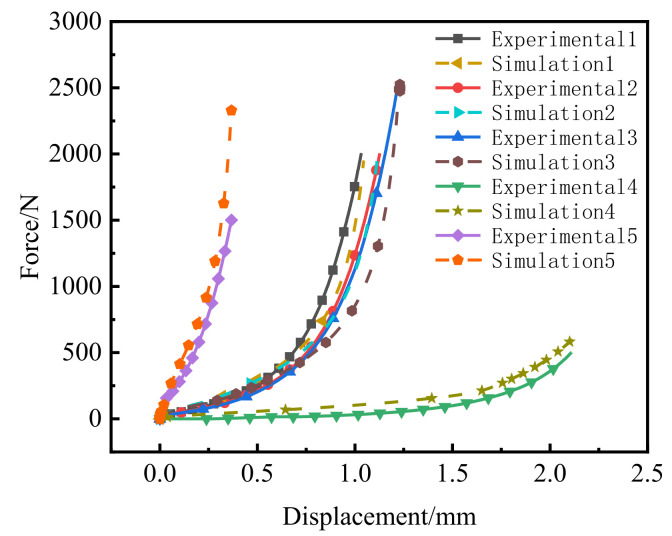
Comparison of the influence of different initial relative densities on molding force.

**Figure 11 polymers-14-01270-f011:**
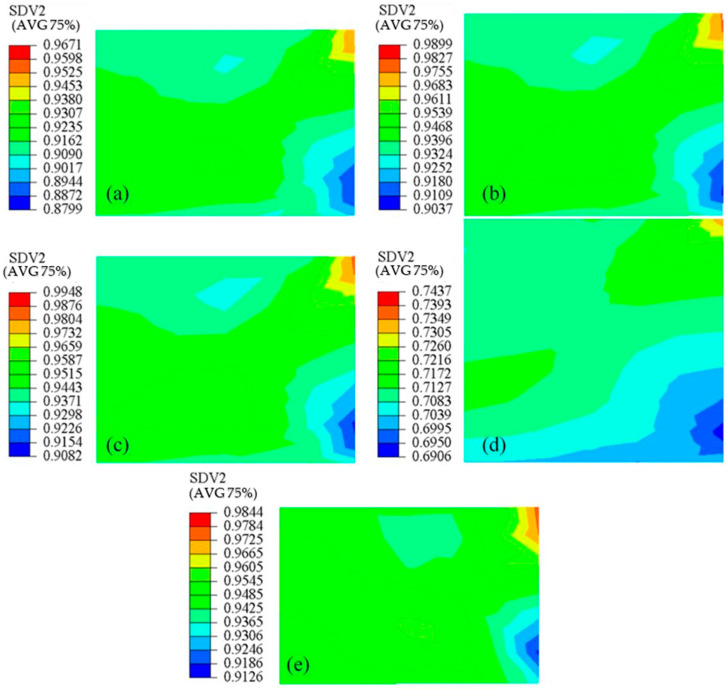
Relative density (SDV2) distribution of powder compact based on simulation results: (**a**) $\rho_{0}=0.4352$; (**b**) $\rho_{0}=0.4485$; (**c**) $\rho_{0}=0.4377$; (**d**) $\rho_{0}=0.2558$; (**e**) $\rho_{0}=0.6767$.

**Figure 12 polymers-14-01270-f012:**
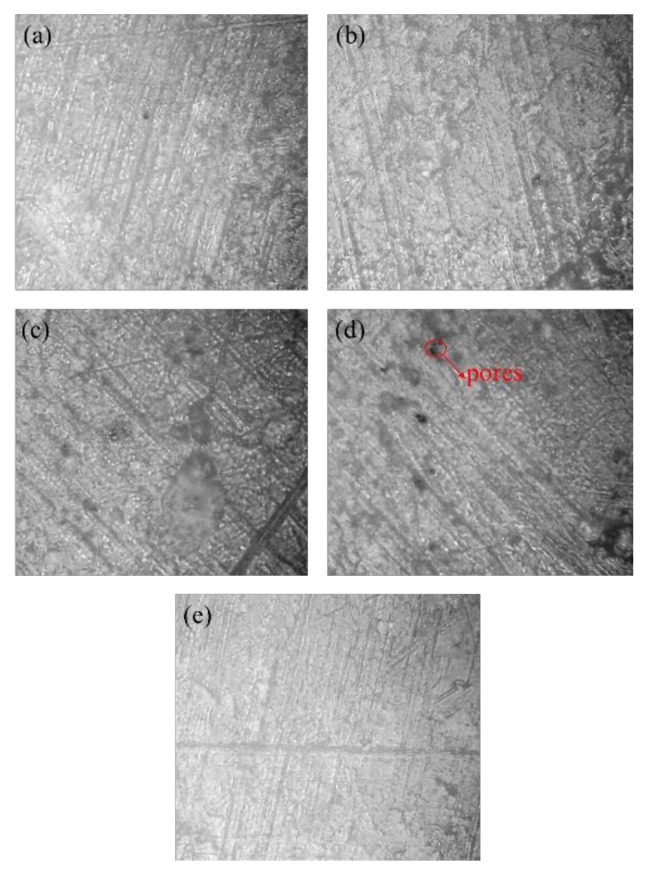
Microstructures at the upper surface of the PEKK compacts at various initial relative densities: (**a**) $\rho_{0}=0.4352$; (**b**) $\rho_{0}=0.4485$; (**c**) $\rho_{0}=0.4377$; (**d**) $\rho_{0}=0.2558$; (**e**) $\rho_{0}=0.6767$.

**Table 1 polymers-14-01270-t001:** Experimental data of green compact.

Num.	Quality/mg	Initial Relative Density	Density	Compaction Terminal Force/N
1	5.44	0.4352	0.9415	2000
2	6.22	0.4485	0.9719	2000
3	6.37	0.4377	0.9837	2500
4	5.56	0.2558	0.7165	500
5	5.38	0.6767	0.9667	1500

## Data Availability

Data sharing is not applicable to this article.

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
