# Peer review of "A Density-Dependent Modified Doraivelu Model for the Cold Compaction of Poly (Ether Ketone Ketone) Powders"

_polymers, 2022, doi:10.3390/polym14061270_

Round 1
Reviewer 1 Report
The manuscript is well organized and overall, well written. The scientific content is most of the time well described and the findings can be useful for many researchers and companies interested in this technology. Therefore, I believe the manuscript could be published.
Author Response
Thank you for agreeing with my article.
Reviewer 2 Report
Dear Authors,
You provided a detailed description of how the research was done.
Advantages and disadvantages of the Euler integration method (critical approach) in the context of the step-by-step method used by the Authors and the characteristics of the differences between the methods were also described favorably from the point of view of the quality of the article.
In the article, You compile the results of real tests and their simulations - I agree that this is the correct research path that allows you to save time and conduct ineffective tests, but I suggest that you need to complete the information on real tests and the starting material that forms the basis of the simulation .
The article lacks precise information and characteristics of the base material on which the simulations are based.
The research process, the degree of compaction, and the pressure effect are described, but the information on the materials was described in a 'negligible' (sparse) manner.
The article is interesting and beneficial from a scientific and practical point of view, but after supplementing the information about the base material (in the form of photos, test results, e.g. density or strength, because this test is described along with simulation or even photos of the microstructure of the working material.
Thank You for Your work and Best Regards,
Reviewer
Author Response
Thanks for your valuable comments. we have made some modifications on the paper. We have added some test results to the draft and discussed these results. We believe these test results will significantly improve the overall quality of this paper.All the changes in the draft make use of the revision function, and the font is red. We would appreciate the reviewer give a second look. As always, we appreciate your valuable time on reviewing our paper, and all your comments and suggestions will be considered.
